# Near-Fall Detection in Unexpected Slips during Over-Ground Locomotion with Body-Worn Sensors among Older Adults

**DOI:** 10.3390/s22093334

**Published:** 2022-04-27

**Authors:** Shuaijie Wang, Fabio Miranda, Yiru Wang, Rahiya Rasheed, Tanvi Bhatt

**Affiliations:** 1Department of Physical Therapy, University of Illinois at Chicago, Chicago, IL 60612, USA; wangshj1985@gmail.com (S.W.); yiru.emma.w@gmail.com (Y.W.); 2Department of Computer Science, University of Illinois at Chicago, Chicago, IL 60607, USA; fabiom@uic.edu (F.M.); rrashe2@uic.edu (R.R.)

**Keywords:** near-fall, gait-slip, balance loss, deep learning, machine learning

## Abstract

Slip-induced falls are a growing health concern for older adults, and near-fall events are associated with an increased risk of falling. To detect older adults at a high risk of slip-related falls, this study aimed to develop models for near-fall event detection based on accelerometry data collected by body-fixed sensors. Thirty-four healthy older adults who experienced 24 laboratory-induced slips were included. The slip outcomes were first identified as loss of balance (LOB) and no LOB (NLOB), and then the kinematic measures were compared between these two outcomes. Next, all the slip trials were split into a training set (90%) and a test set (10%) at sample level. The training set was used to train both machine learning models (*n* = 2) and deep learning models (*n* = 2), and the test set was used to evaluate the performance of each model. Our results indicated that the deep learning models showed higher accuracy for both LOB (>64%) and NLOB (>90%) classifications than the machine learning models. Among all the models, the Inception model showed the highest classification accuracy (87.5%) and the largest area under the receiver operating characteristic curve (AUC), indicating that the model is an effective method for near-fall (LOB) detection. Our approach can be helpful in identifying individuals at the risk of slip-related falls before they experience an actual fall.

## 1. Introduction

Falls are one of the main causes of fatal and non-fatal injuries among older adults [1]. Approximately one out of three older adults aged over 65 falls each year and this number increases as the population ages [1]. Slip perturbation is considered one of the leading causes of falls in independent community ambulation for older adults [2,3,4,5]. Slip-induced falls can lead to serious consequences including hip/arm fractures, traumatic head injuries, functional and mobility decline, and increased dependency, thereby resulting in long-term disability and even death [6,7,8]. Accurate records of daily fall events could help to identify the population at high risk of falling and ensure that fall prevention interventions are properly provided to targeted individuals, considering that older adults who have a history of falls are more likely to experience recurrent falls [9]. Wearable sensors can provide an objective record of daily activities and improve the efficiency of fall detection; therefore, the fall risk detection approach based on wearable sensors has been the focus of substantial research and is of ultimate importance in fall prevention among the geriatric population [10,11,12].

Accelerometer-based sensors are the most used sensor type in studies on fall detection [10,11,13,14]. Among studies on fall detection, young adults are the major study population reported on, rather than older adults who are more vulnerable and predisposed to a fall [1]. A systematic review paper of sensor application in older adults revealed that most published studies of sensor application among older adults are for fall risk screening [11]. In the former type of studies, sensors are only worn during activities of daily living such as walking [15] or during a baseline clinical assessment such as a balance test or Timed Up and Go test [16], while the fall outcomes in the majority of studies are still recorded prospectively [17,18] or retrospectively as subjective self-reported fall events [19]. Although this type of study is crucial in the early identification of individuals at high risk of falls based on predictive models, the self-reported fall technique adopted in such designs has limitations such as recall bias (bias in questionnaires due to inaccurate recall) [20] and relies on subjects’ motivation [21]. Even though a few studies recorded the fall event during the long-term monitoring of the subjects, the fall events were only collected in specific indoor environments [22,23], while falls mainly occurred outside the home induced by external perturbations [24,25]. Therefore, the study of automatic fall detection using sensors that are worn during fall events is essential in improving alarm systems to identify older adults who need immediate assistance. However, automatic perturbation-induced fall detection studies on older adults are limited [11].

Fall detection based on real-life falls is, when compared to the approach based on laboratory simulated falls, a less heterogeneous approach aimed towards understanding daily fall mechanisms [26]. However, recording real-world falls is both costly and time consuming due to the rarity of the event [27], and hence the application of this method is limited. Only 7.1% of sensor-based fall detection studies reported monitoring study participants in a real-world setting [28]. Therefore, the use of laboratory simulated falls has been a cost-effective and safe supplement to understand the efficacy of sensor-based real-world falls detection and is adopted by a vast majority of studies [10,28]. It has been demonstrated that real-life forward falls, sideway falls, and backward falls have similar features to those from simulated falls [26]. Furthermore, a previously developed model based on simulated falls showed a high accuracy (>80%) of fall-related abnormal gait pattern classification for young subjects (<50 years) [29]. However, the study of wearable-sensor-based fall detection on older adult populations using simulated falls is still in its infancy and only a limited number of studies appeared in the authors’ literature search.

Laboratory-reproduced falls through unexpected perturbations could reduce the unpredictability of intentional simulated falls and be combined with body-worn sensors to examine fall detection. Perturbations are introduced by unexpected postural disturbance to simulate the accidental nature of falls [30]. For safety reasons, falling is terminated by protective devices such as a protective harness system, which prevents the impact on the ground, and the changes of motion due to a fall accident can be examined in the near-fall period before a fall occurs. A near-fall could be defined as a loss of balance (LOB) which initiates a falling but does not result in a fall to the ground or lower surfaces [31]. A few studies have found near-falls to be more frequent than actual falls and a clinically relevant markers of falls [31,32,33,34,35]. Given the ability of near-fall detection to identify older adults at a high risk of falling before a devastating fall event [36], sensor-based objective techniques which could quantify near-fall events are worth further study.

To our best knowledge, only two studies have applied unexpected perturbations on older adults to investigate sensor-based near-fall detection [37,38]. Both studies, however, focused on analyzing trips during treadmill perturbations. This presents a major limitation, considering that differences exist between overground walking and treadmill walking in healthy older adults—older adults tend to have greater cadence, a smaller stride length and stride time as well as reductions in the majority of joint angles, moments, and powers in treadmill walking [39]. Considering that (1) the slip outcomes for the same individual might be different with regard to overground walking and treadmill walking and (2) slip-related falls comprise around 40% of outdoor falls among older adults [40], near-fall detection methods induced by slips during overground walking must be investigated.

Threshold-based algorithms and intelligent algorithms are the two main methods in fall detection using wearable devices [41,42]. Threshold-based techniques can distinguish different classes based on the magnitude of accelerations; such techniques have low computational complexity and are easy to apply. However, this method is only suitable for specific types of falls and cannot be applied to individuals who differ in weight and height [43]. Hence, intelligent detection algorithms, which can automatically extract features related to falls or near-falls, have been increasingly used for fall detection in the past decade [44]. Intelligent detection algorithms include machine learning methods and deep learning methods. Machine learning is closely related to computational statistics, which primarily focuses on classification and regression based on known features previously learned from the training data. Deep learning allows for the creation of computational models that are composed of multiple processing layers and learn representations of data with multiple levels of abstraction, which is a more powerful and efficient way to deal with massive amounts of data [45]. In this study, we applied and compared the usage of different methods for near-fall detection over different cases. The goal of this study was to investigate whether the near-fall event (LOB) following an unexpected perturbation in overground walking could be accurately identified based on body-fixed sensor-collected accelerometry data. In addition, the performance of traditional machine learning methods and deep learning approaches were compared for the slip outcomes classification.

## 2. Methods

### 2.1. Participants

Thirty-four community-dwelling older adults (≥60 years) participated in this study. All participants were initially screened to pass a cognitive test (>25 on the Folstein Mini Mental Status Exam) [46], a calcaneal ultrasound screening (T-score > −2.0) [47], a mobility test (Timed Up and Go < 13.5 s) [48], and a monofilament foot sensation test (able to detect the Weinstein 5.07 monofilament at all nine locations on both feet) [49]. Exclusion criteria included recently (≤6 months) self-reported diagnosed neurological, musculoskeletal, or other systemic disorders. All participants provided written informed consent, and this study was conducted according to the guidelines of the Declaration of Helsinki and approved by the Institutional Review Board of the University of Illinois at Chicago (IRB#: 2016-0887).

### 2.2. Experimental Setup

The slip perturbation was induced by releasing a low-friction, movable platform embedded near the middle of a 7-m walkway. The platform was firmly locked in the first ten walking trials. During the slip trial, the platform could slide freely in the anteroposterior (AP) direction for up to 60 cm forward or 10 cm backward. Once a subject’s right (slipping) foot was detected in contact with the right platform by the force plates (AMTI, Newton, MA, USA) installed beneath the platforms [50], a computer-controlled triggering mechanism would release the platform. Participants were instructed to walk at their preferred speed and in their preferred manner, and they were told that a perturbation may or may not happen during any of the trials. All participants experienced 24 unexpected slips and around 20 unperturbed trials in between these to reduce the participants’ anticipation of the slip perturbation. The duration of each trial was 10 s, which was from gait initiation to gait termination on the 7-m walkway. Only the slip trials were analyzed for this study. In total, 816 slip trials were collected for the 34 participants, while 18 slip trials (2%) were excluded due to data collection issues (e.g., dropped marker or moved sensor). Therefore, 798 slips were analyzed in this study.

The participants wore their own athletic shoes and a full-body safety harness connected with shock-absorbing ropes to a loadcell (Transcell Technology Inc., Buffalo Grove, IL, USA) mounted on an overhead trolley on a track over the walkway, enabling participants to walk freely while providing protection against body impact with the floor surface. A 3-axis accelerometer, Axivity AX3 (Axivity Ltd., Newcastle, UK), was attached on the lower back of the participants using an elastic belt; the accelerometer registered 3D sacrum acceleration at 100 Hz and ±8 g. To identity the slip outcomes and compare the reactive performance between different slip outcomes, kinematics of a full body marker set of 30 retro-reflective markers were also recorded by an eight-camera motion capture system (Motion Analysis Corporation, Santa Rosa, CA, USA). Kinematic data were sampled at 120 Hz and synchronized with the loadcell data and force plate data, which were collected at 600 Hz. It should be noted that the camera system was only used for the model development but not for its application.

### 2.3. Slip Outcomes and Reactive Kinematics

Slip outcomes were classified as LOB or NLOB. LOB was defined as the recovery foot landing posterior to the sliding foot based on the location of heel markers [51]; otherwise, the slip outcome would be NLOB. It is known that the slip severity or slip consequence is related to the slip outcomes. To have a better understanding of the difference in the consequences between the two slip outcomes, we compared the reactive performances of participants with respect to the LOB and NLOB trials, including the recovery stride length, slipping distance at recovery foot touchdown (TD), the slipping velocity at TD, and trunk extension angles. These variables were calculated from the motion data in the slip trial. The recovery foot is the one taking a compensatory step following slip perturbation, and TD is the instant when the recovery foot contacts the ground, which was detected from force plate data. Recovery stride length was calculated as the travel distance of recovery heel from slip onset to TD in AP directions. The slip distance and velocity were approximated by the slider marker. The trunk angles in sagittal plane were calculated as the angle between vertical lines and the straight line connecting the center of hip markers and the center of shoulder markers.

### 2.4. Classification Models

Although the traditional threshold-based classification method showed a high accuracy (>80%) for fall detection [27,52,53], the previously reported threshold of 3.52 g (corresponding to 2.52 g after gravity removal in our study) for vertical acceleration failed to distinguish LOB and NLOB trials in our study. Take the trials in Figure 1, for example; the LOB trial showed an immediate increased acceleration in all the three axes within 1 s following the perturbation due to the trunk reaction. For the NLOB trial, the acceleration was also increased in all the three axes at around 1 s after the slip onset, which might be related to the stepping of the trailing limb. For this case, only the acceleration in a mediolateral direction was larger in the LOB trials compared to the NLOB trials. Vertical acceleration was similar in both, and the AP acceleration was even smaller in the LOB trials. Furthermore, both trials exceeded the fall threshold due to the unexpected slip perturbation, indicating that the threshold-based method is not suitable for slip outcomes classification.

Therefore, machine learning and deep learning models were used to identify the slip outcomes based on the 10 s time series acceleration data for each trial (from gait initiation to gait termination). Specifically, two machine learning methods were used: time series forest classifier (TSF) [54] and time series classification with multiple symbolic representations and symbolic sequence classifier (Mr-SEQL) [55]. TSF is a tree-ensemble method proposed for time series classification. It uses an ensemble of decision trees (i.e., random forest) that use both entropy and distance to capture temporal characteristics. Mr-SEQL uses a symbolic sequence learning algorithm to efficiently traverse the feature space and select the most discriminative subsequences of the time series for a linear model.

On top of that, we also used two state-of-the-art deep learning approaches, time Le-Net (TLeNet) [56] and InceptionTime (Inception) [57]. TLeNet is a deep learning approach that uses a data-augmentation technique and convolutional neural network consisting of two convolutional layers. The state-of-the-art Inception model is an ensemble of five different deep learning models for time series classification; each model has the same architecture but with different randomly initialized weight values.

K-fold (k = 10) cross-validation was used to calculate the performance of all the models. The slip trials were randomly split into a training set (90%) and a test set (10%) at sample level; therefore, the rate of LOB and NLOB was similar in both datasets. The training set was used to train the model parameters and evaluate the resulting models by minimizing the loss, and the test set was used to evaluate the performance of each model. Considering the fact that participants could adapt to the slip perturbation [58], there were more NLOB data points than LOB. Because of this proportion problem, we employed ADASYN, an oversampling approach used for learning from imbalanced data sets [59]. The model training and evaluation were performed using the Python 3 libraries sktime and sktime-dl.

### 2.5. Statistical Analysis

To evaluate the performance of each model for slip outcome classification, first of all, the receiver operating characteristic curve (ROC) was calculated. Following conventional ROC analysis, the thresholds of the ROC plot were determined by sorting the test instances decreasing by target scores, and processing one instance at a time to update the TP and FP. Next, the sensitivity for LOB detection [TP/(TP + FN)], the specificity for NLOB detection [TN/(TN + FP)], and the overall classification accuracy was calculated for each model using three different approaches: (1) default cutoff at which the overall accuracy is maximum; (2) optimal cutoff at which the sum of sensitivity and specificity is maximum; and (3) default cutoff along with the ADASYN oversampling approach. The area under the curve (AUC) is a robust metric of a model’s performance; hence, the AUC of each model was also calculated for each test set (*n* = 10), and one-way ANOVA was conducted to compare the difference in AUC among these four models. The post-hoc paired *t*-test was then used to compare each pair of them. All statistical analysis was performed using MATLAB2021a (MathWorks Inc., Natick, MA, USA).

## 3. Results

Among 798 slip trials, 229 of them were identified as LOB (40 are falls), and the rest (569) were classified as NLOB. The participants with LOB showed a significant difference in terms of reactive kinematics compared to those without LOB (*p* < 0.001 for all; Table 1). The LOB trials had longer slip distances (31 vs. 8 cm) and faster slip velocity (1.12 vs. −0.34 m/s) than the NLOB trials at recovery touchdown, which resulted in more trunk extension (4.9° vs. −1.46°) and a shorter recovery stride length (0.41 vs. 0.70 × body height) for the participants losing their balance.

The time series acceleration data could be used to identify the slip outcomes based on all the models. Specifically, the machine learning models (TSF and Mrseql) showed a lower classification accuracy (<81%, Table 2), and the two deep learning models (TLeNet and Inception) showed a higher classification accuracy (>82%). Among all the models, the Mrseql model showed the lowest overall accuracy, with 69.5% at the default cutoff, and the Inception model showed the highest overall accuracy, with 87.5% at the default cutoff. Specifically, this model could accurately identify 94.8% of NLOB, and 69.4% of LOB.

The models at the optimal cutoff (Figure 2) were highly consistent with those at the default cutoff. Among them, the Inception model still showed the highest overall accuracy (85.2%) and the identification accuracy for LOB increased to 86.5%, while the accuracy for NLOB reduced to 84.8% (Table 2). For the models using the ADASYN approach, the sensitivity for all the models were improved by ~5%, while their specificity was reduced, except in the case of the TLeNet model. Therefore, the overall accuracies were similar between the models with and without the ADASYN approach.

The one-way ANOVA results showed that method had a significant effect on the AUC (F = 43, *p* < 0.001, Figure 3). Post-hoc tests indicated that the Inception model had the largest AUC compared to the other three models (*p* ≤ 0.01 for all), while the Mrseql model showed the smallest AUC among all the models (*p* < 0.001 for all). The TSF and TLeNet models showed a comparable AUC (*p* > 0.05).

## 4. Discussion

Our results indicated that machine learning models could accurately (>82% for TSF and Mr-SEQL) identify the NLOB trials, while they could only identify the LOB trials with a low accuracy (<50% at default cutoff). The deep learning models (TLeNet and Inception) showed a higher accuracy for both LOB (>64%) and NLOB (>90%) classifications, indicating that deep learning is a more powerful method for near-fall (LOB) detection.

Compared to participants in the NLOB trials, participants with balance loss always experienced a longer slip displacement (31 ± 16 cm) and faster slip velocity (1.12 ± 0.74 m/s). Previous studies have showed that once the slip displacement exceeds 30 cm, the probability of a fall is around 50% [60]. A faster slip velocity further increases the fall risk to around 70% once the slip velocity relative to COM exceeds 1 m/s [61]. Therefore, even if individuals with LOB prevent a fall by taking a reactive (backward) step, they are still at a higher fall risk compared to those without LOB. Furthermore, it has been reported that the slip outcome (fall or not) following LOB is determined by the performance of reactive stepping [62], as reactive stepping can restore the stability and enhance limb support to avoid a fall [63]. However, reactive responses can be influenced by many factors, such as muscle fatigue, improper landing location, or longer reaction time [62,64,65]. It is possible that individuals who prevent a fall following LOB in one slip might experience a fall when they encounter another unexpected slip. Hence, individuals with LOB should be detected and receive fall intervention or balance training to lower their risk of slip-induced falls [66,67]. Additionally, participants with LOB showed a larger trunk extension following the slip perturbation, which might induce trunk injury or back injury [68]. Quick stepping with a shorter stride length (or larger distance between the two feet) would be required for balance recovery, which might induce muscle strain. Therefore, near-fall events could also cause injuries and should be detected.

Several studies have tried to detect fall events arising during activities of daily living [53,69,70,71,72], which are voluntary movements; motor responses following slip perturbation, on the other hand, are reactive movements. Previous studies have compared reactive stepping and voluntary stepping and revealed that there were significant differences in the locomotor performance between these two behaviors [73], for example, in terms of step length, execution time, joint angles, and different neuromuscular responses. Due to these differences, fall events can be accurately detected based on accelerometry data using the traditional threshold-based method; once the magnitude of acceleration in an event exceeds the determined threshold, the event would be identified as a fall. Otherwise, it would be a non-fall. Contrary to previous studies, this study aimed to distinguish between participants with and without LOB that experienced a slip perturbation under similar conditions, to elicit recovery reactive responses. In this case, the acceleration showed similar magnitudes in the LOB and NLOB trials (Figure 1); therefore, the traditional threshold-based method would not be suitable. Even the machine learning methods selected in this study (TSF and Mr-SEQL) only showed a low classification accuracy for LOB detection due to their similarity.

Compared to machine learning models, deep learning models showed a higher classification accuracy, especially the Inception model, which had the highest classification accuracy and the largest AUC. Such results indicate that the Inception model is a more powerful wayto identify reaction characteristics based on accelerometry data. Previous approaches, including TLeNet, were inspired by image recognition and made use of progressive pooling layers to reduce the input data’s dimensionality. This model negatively impacted accuracy since it ignored valuable information in favor of a simpler model. Conversely, fully convolutional neural networks were shown to achieve better performance without pooling layers. Considering this, and the fact that time series are dimensionally simpler than images, it became possible to design more complex models, such as Inception. This model, by applying multiple filters simultaneously to the input time series, allows the network to extract relevant features considering time series of different lengths, which is an essential characteristic used to capture the subtle acceleration changes in our data (Figure 1).

Although the Inception model showed a high accuracy (94.8%) for NLOB detection at the default cutoff, it could only detect the LOB with an accuracy of 69.4%. One of the reasons for this problem is the imbalance between the LOB and NLOB groups. Our results indicated that the ADASYN approach could lower the effect of imbalanced sample sizes on the classification accuracy. Additionally, the harness system used in this study might be another factor affecting the classification accuracy, as the impulse-absorbed harness system limited the changes in the vertical acceleration following the slip perturbation. Furthermore, there was no typical impact stage for the LOB trials due to the protection of the harness system. While the sudden deceleration during the impact stage is a key characteristic for fall detection, it is reasonable to postulate that the classification accuracy of LOB (including both falls and non-falls) would be higher in the home environment without using the harness system.

To the best of our knowledge, this study is the first to identity slip-induced near-falls based on acceleration data. Previous studies only developed models for slip event detection without differentiating LOB and NLOB events [74,75]. Furthermore, the reliability of these models needs to be further validated, as only small sample sizes (<9) were used for validation. Compared to other models for fall risk classification based on accelerometer data and/or gyro data [76,77,78], our model showed a comparable accuracy (>80%). It should be noted that these previously developed models only distinguished fall events from activities of daily living (ADLs), and most of the ADLs could only generate an acceleration with a small magnitude; thus, the inclusion of a large number of ADLs would greatly increase the identification accuracy. As aforementioned, our study only included slip trials with a similar performance, and such a method of selecting a dataset made the development of our model more challenging but also made the model more reliable. Additionally, there are some state-of-the-art fall detection algorithms with a higher (>90%) prediction accuracy [79,80], fusing accelerometer data with gyro data. This indicates that fusion data might further improve the fall/near-fall risk classification accuracy. Hence, our future study will try to enhance our model based on fusion data.

Limitations exist in this study. First, only slip-induced LOB trials were analyzed; trip perturbation is another major cause of falls in older adults [81,82]. In contrast to slip perturbations, trip perturbations can cause a forward LOB. Due to the difference in the motor behaviors, different models might be developed for trip-related LOB detection. Furthermore, we only tested the developed models using perturbed walking trials, and the question of whether the models can accurately classify conventional ADLs without perturbation (i.e., normal walking) remains unclear. However, a previous study developed a model to detect slip events during normal walking based on an inertial sensor [74]. The results indicated that slip walking showed different angular and heel accelerations compared to normal walking, and the slip event could be accurately classified from the walking trials. Hence, this study only focused on the most challenging problem by including only perturbed trials. Our future study will try to collect a diversity of conventional ADLs to further validate the reliability of our model.

## 5. Conclusions

In conclusion, this study developed a near-fall detection model based on a wearable accelerometer for older adults, and our results indicated that a deep learning method could accurately classify LOB and NLOB events following a slip perturbation. This near-fall detection approach could be used to identify individuals at risk of slip-related falls before they experience an actual fall, which might contribute to reducing slip-related injuries.

## Figures and Tables

**Figure 1 sensors-22-03334-f001:**
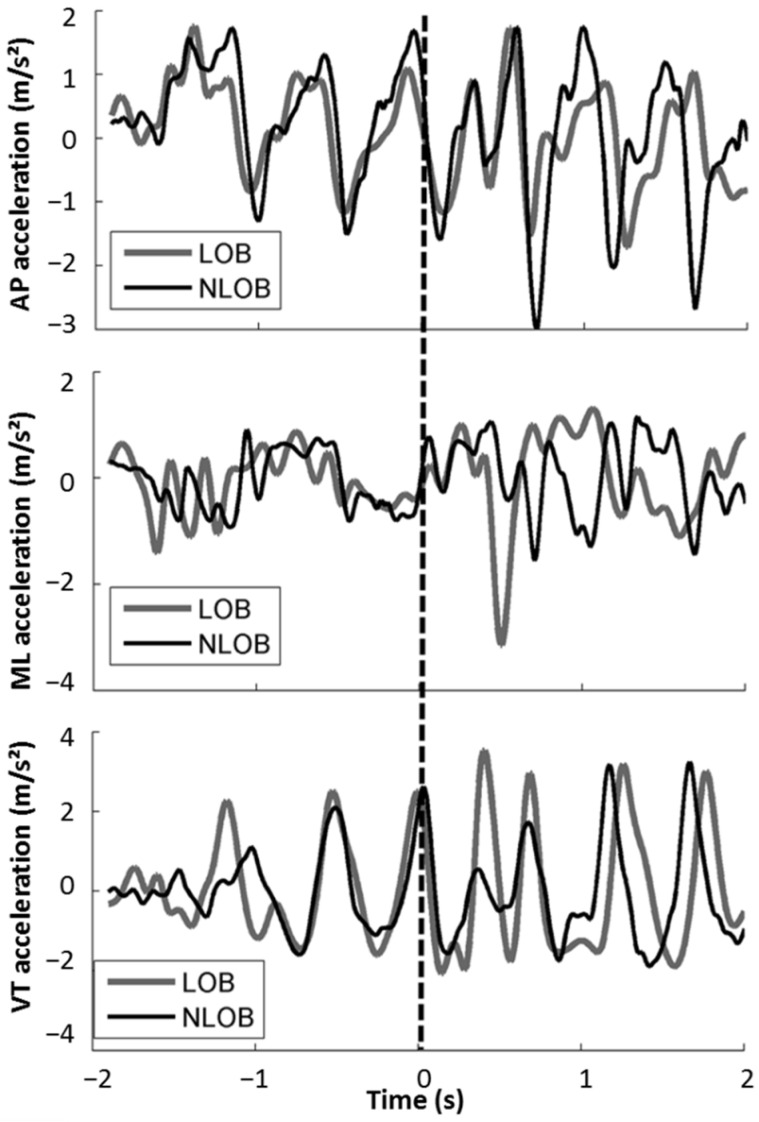
One sample of three-axis acceleration time-series curves for an LOB trial (gray) and an NLOB trial (black), the time of 0 s indicates the onset of slip perturbation. AP indicates anterior-posterior direction, ML indicates mediolateral direction, and VT indicates vertical direction.

**Figure 2 sensors-22-03334-f002:**
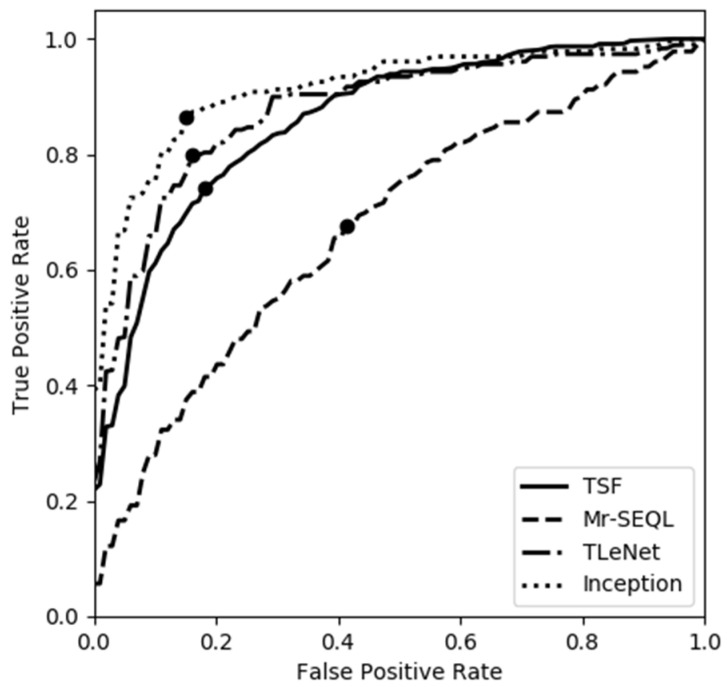
The receiver operating characteristic curve (ROC) for the machine learning (TSF and Mr-SEQL) and deep learning (TLeNet and Inception) models. The true positive rate is the sensitivity, and the false positive rate is 1- specificity. The optimal cutoff is shown as a circle for each model. Among all these models, the Inception model showed the highest specificity and sensitivity, followed by the TLeNet and TSF models, while the Mr-SEQL model showed the worst performance.

**Figure 3 sensors-22-03334-f003:**
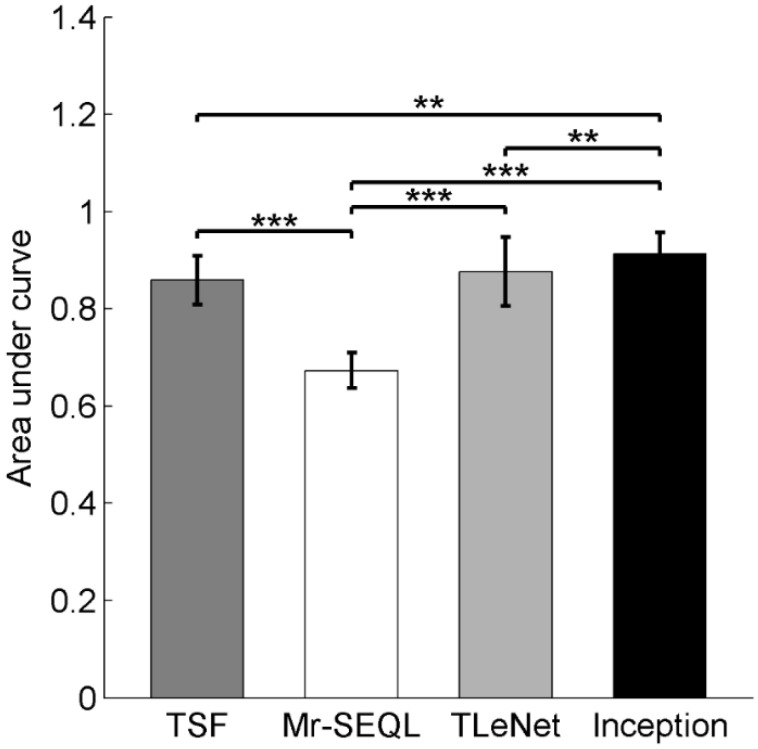
Post-hoc comparison of the area under curve (AUC) across the TSF, Mr-SEQL, TLeNet, and Inception models. The Inception model had a significantly larger AUC value than the other three models, and the Mr-SEQL model had a significantly smaller AUC value than other models. ** indicates <0.01, *** indicates <0.001.

**Table 1 sensors-22-03334-t001:** Comparison of kinematic measures (mean ± standard deviation) between LOB trials and NLOB trials.

Method	LOB	NLOB	*p* Value
Stride length/height	0.41 ± 0.18	0.70 ± 0.15	<0.001
Slip distance (m)	0.31 ± 0.16	0.08 ± 0.13	<0.001
Slip velocity(m/s)	1.12 ± 0.74	−0.34 ± 0.62	<0.001
Trunk angle(degree)	4.9 ± 8.22	−1.46 ± 7.72	<0.001

**Table 2 sensors-22-03334-t002:** Specificity (for NLOB), sensitivity (for LOB), and overall classification accuracy of the two-class models at default cutoff and optimal cutoff, as well as the results using the adaptive synthetic sampling approach (ADASYN). Spe indicates specificity, Sen indicates sensitivity.

Method	Accuracy at Default Cutoff			Accuracy at Optimal Cutoff			Accuracy with ADASYN		
	Spe	Sen	Overall	Spe	Sen	Overall	Spe	Sen	Overall
TSF	94.4%	45.4%	80.3%	81.8%	74.3%	80.0%	86.1%	56.8%	77.6%
Mr-SEQL	82.7%	37.1%	69.5%	58.6%	67.7%	60.8%	73.7%	40.7%	64.1%
TLeNet	90.2%	64.2%	82.7%	83.8%	79.9%	82.9%	90.4%	67.7%	83.8%
Inception	94.8%	69.4%	87.5%	84.8%	86.5%	85.2%	92.0%	73.8%	86.7%

## Data Availability

The data presented in this study are available on request from the corresponding author. The data are not publicly available due to privacy issue.

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
