# Peer review of "Near-Fall Detection in Unexpected Slips during Over-Ground Locomotion with Body-Worn Sensors among Older Adults"

_sensors, 2022, doi:10.3390/s22093334_

Round 1

Reviewer 1 Report

The paper describes wearable sensor-based detection and classification of slip events during overground walking. The classification aims to distinguish between slips that determine a fall and non-fall related slips and adopts machine learning and deep learning to do so, comparing the results from various methodologies. The paper itself touches an important topic and describes an interesting research idea but of course there is room for improvement. A list of possible issues and question follows, hoping to contribute improving this paper.

Reviewer 2 Report

The paper is interesting. However, I think that authors should revise it and address the following concerns:

It is not true that the all studies regarding fall detection systems and elderly population “fall outcomes are still recorded prospectively 9 or retrospectively by subjective self-re-44 ported fall event”. There are some works (and even datasets) based on falls collected during a long-term monitoring of older adults. See for examples, the studies related to FARSEEING or FFFStudy dataset, or the article by Harari et al.:

‘A smartphone‑based online system for fall detection with alert notifications and contextual

information of real‑life falls’

This point should be discussed in the paper.

The fact that induced falls (or near falls) behave in the same way as ‘spontaneous’ real falls (or near falls) is still a controversial issue in the related literature. Although this topic is addressed in the paper, in my opinion, it should be discussed in more detail. There are some missing works that have utilized induced near-falls. See, for example: ‘A baseline walking dataset exploiting accelerometer and gyroscope for fall prediction and prevention systems’, by Masoud Hemmatpour et al.

In a real application scenario of a movement analyzer as that proposed in the paper, the system should be capable of discriminating LOB actions from any other ADLs (not only from other similar movements, collected during the same actions, which did not produce a LOB). Have you tested the classifier when it is inputted with the acceleration signals collected from a diversity of conventional ADLs?

Authors should analytically describe how the parameters employed to characterize the movements (e.g. recovery stride length, slipping distance at recovery foot touchdown (TD), the slipping velocity at TD, and trunk extension angles) have been computed from the acceleration or force plate data.

The inputs with which the models are fed are not completely explained. Which data are really inputted to the classifiers? 10-s of the tri-axial components of the captured acceleration? 3000 values (as the signal has three ‘channels’ and the sampling rate is 100 Hz)?  A trip normally lasts much less than 10 seconds? Why is this observation window selected?

I strongly recommend to make publicly available the interesting dataset that has been collected for this study.

Slip is not the only cause of falls or loss of balance. Why not considering tripping/other types of stumbling?

The receiver operating characteristic curve (ROC) and the corresponding AUC metric are computed by modifying a certain thresholding or decision criterion. Which value have you modified in the employed classifiers?

The achieved discrimination ratio are in general a bit poor (with sensibilities and specificities lower than 87% for all the models). Compare this results with those obtained by other similar results of the state-of-the-art.

In a real scenario, an operating system should be trained/configured with the data collected from an experimental user different from those who will wear the final trained detector. So, I wonder if the classifier works OK if the subset of samples used for testing is different from that used for training.

Typo.

Line 142. Unfinished sentence: ‘Slip outcomes and reactive kinematics’

Minor aspects:

Why using the same number for two different references? E.g. ref 43(a) and 43 (b)

Round 2

Reviewer 1 Report

The authors carefully reviewed the paper after the first round, improving its overall quality and making it acceptable in its present form.

Reviewer 2 Report

Authors have improved the draft and, in general, have addressed and discussed my previous concerns.